# GLARE: Towards Graph-less Retrieval for Retrieval Augmented Generation on Million-scale Knowledge Graphs

## Abstract

Retrieval-augmented generation (RAG) has emerged as an effective solution to mitigate hallucinations in Large Language Models (LLMs) by retrieving from an external knowledge base. Recent works have explored KG-based RAG, which leverages knowledge graphs (KGs) to incorporate rich relational information. However, existing methods suffer from high retrieval latency, as the retriever model needs to directly operate over graph space, thereby hindering their scalability to large-scale KGs. We propose GLARE, a scalable KG-based RAG framework that enables fast and accurate information processing over million-scale KGs. Specifically, GLARE compresses large KGs into a compact, knowledge-intensive vector memory, enabling efficient retrieval without searching over an exponentially vast graph space. To preserve critical information, we further design a non-parametric, importance-aware graph pooling strategy and a VAE-style projector that reconstructs relational structures from the vector memory. At inference time, GLARE enables linear-time retrieval from the vector memory, significantly accelerating KG-based question-answering (QA) while maintaining high response quality. Evaluations on the STaRK benchmark across multiple domains demonstrate that GLARE achieves over $\times 30,000$ retrieval speedup with improved question-answering performance.

## 1 Introduction

The retrieval-augmented generation (RAG) system (Lewis et al., 2020) has emerged as a prominent approach for incorporating additional knowledge sources and reducing hallucinations in Large Language Models (LLMs) (Kaddour et al., 2023; Zhang et al., 2023b; Huang et al., 2025). Traditional RAG systems typically enhance LLMs with an external vector-indexed database, where both documents and input queries are embedded into a shared vector space using an encoder. At inference time, relevant documents are retrieved based on vector similarity and incorporated into the LLM's prompt. Thus, LLM can utilize external knowledge to improve answer factuality and reduce hallucination (Liu et al., 2025a).

There is a recent surge of interest in exploring the RAG systems enhanced by knowledge graphs (KGs) (Ji et al., 2022), where structured knowledge is employed as an external knowledge source to support LLMs (Pan et al., 2024). Compared with vector knowledge bases in traditional RAG, KGs offer a more flexible alternative to aid LLMs with rich external knowledge (Huang et al., 2025), particularly advantageous in two aspects. First, they explicitly capture relational structures between entities, which are essential for multi-hop reasoning across interconnected facts (Yang et al., 2024; Liu et al., 2025b). Second, KGs can integrate heterogeneous data sources, enabling complex reasoning across diverse knowledge domains (Wu et al., 2019). Due to the advantages of KGs, KG-based RAG has the potential to facilitate LLMs with many crucial applications, such as medical diagnoses (Su et al., 2025; Dou et al., 2025), scientific discovery (Gao et al., 2022; Cheng et al., 2025), misinformation mitigation (Zhang et al., 2023a; Li et al., 2025b), and finance (Peng et al., 2024a). However, a major bottleneck of KG-based RAG lies in inefficient knowledge retrieval (Peng et al., 2024b; Han et al., 2024). KGs are highly irregular and discrete graphs, where the entities in KGs are interconnected through their relationship. The knowledge retrieval process of KG-based RAG needs to search over the graph structure to find a desired substructure, such as an entity, a

path, or a subgraph, that contains relevant knowledge to the input query. Thus, the searching space increases *exponentially* as the size of KG grows (Chen et al., 2024), making the knowledge retrieval of KG-based RAG computationally expensive (Peng et al., 2024b; Han et al., 2024).

Prior work on KG-based RAG has focused on the design of efficient retrieval models for KG-based RAG by leveraging Graph Neural Networks (GNNs) and Large Language Models (LLMs) (Kim et al., 2023; Gutierrez et al., 2024a; Ye et al., 2022; Mondal et al., 2024; Tang et al.). GNN-based retrievers learn the importance of nodes or subgraphs within the knowledge graph for the question answering (QA) task. At inference time, the contents of the retrieved substructures are used to facilitate the QA task with LLMs. Alternatively, LLM-based retrievers adopt iterative retrieval strategies by exploring multi-hop traversal over the KG to collect relevant information. However, these methods can only handle small-scale KGs with around one thousand entities (i.e., *thousand-scale KGs*) (Sun et al., 2024; He et al., 2024). When applied to large-scale KGs, they still suffer from significant retrieval latency and struggle to pinpoint the relevant information from KGs, as their retrieval process must still operate on a vast and discrete graph space (Ji et al., 2022; Meduri et al., 2024; Hambarde & Proença, 2023; Cui et al., 2023).

We take an orthogonal approach to address the scalability issue by making the retrieval process of KG-based RAG **graph-less**. Specifically, we propose **GLARE**, which compresses the KG into a high information-intensive vector memory. As shown in Figure 1, the retriever model only needs to retrieve from the vector memory at the cost of linear-time retrieval complexity instead of exploring the vast graph space in KGs. Thus, GLARE can scale up KG-based RAG to large-scale KGs with more than one million entities (*million-scale KGs*). When building the vector memory, GLARE employs importance-aware graph pooling to fully utilize the relational information from KGs. This approach identifies the influential nodes in KGs as the anchor points for graph pooling. The neighbor subgraph centered at the influential node is pooled into a knowledge-intensive vector. In this way, the information in the neighborhood around an identified influential node is compressed into a single vector in the vector memory, improving knowledge intensity while reducing information redundancy. To decode the information of the vector memory into LLM-understandable knowledge, we employ a VAE-based objective (Kingma et al., 2013) to train a lightweight projector. This projector maps the message in vector memory

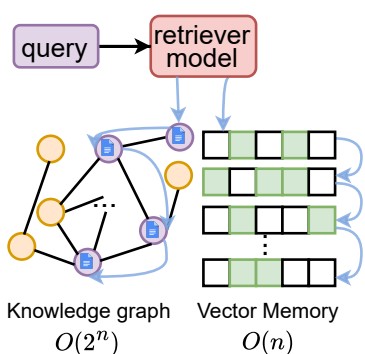

Figure 1: Comparison between knowledge-graph based retrieval and vector memory-based retrieval. The time complexity of retrieval in knowledge graph is $O(2^n)$ (Yu et al., 2021), while in vector memory is $O(n)$.

into the token space of the frozen LLM, teaching the LLM to rephrase the textual information and better recover the structural information in the original KGs. With the projector, the LLM can utilize the vector memory as a 'cheat sheet' to address new queries at inference time.

We extensively evaluate GLARE on the STaRK benchmark (Wu et al., 2024), which involves KG-based knowledge-intensive question-answering (QA) in e-commerce, clinical, and scientific domains. The STaRK benchmark challenges KG-based RAG with retrieval from large-scale KGs, long text documents, and open-ended QA. Extensive experiments show that compared with existing KG-based RAGs, GLARE enjoys over $\times 30,000$ retrieval speedup with competitive QA performances.

Our contributions are summarized as below:

- We propose GLARE, a novel framework that significantly enhances the scalability of KG-based RAG. Instead of optimizing retrieval models over KGs, GLARE compresses the entire KG into a compact, information-intensive vector memory. Thus, GLARE scales up KG-based RAG to million-scale KGs by efficient retrieval without traversing the graph.

- To make the vector memory interpretable to large language models (LLMs), we train a lightweight projector using a variational autoencoder (VAE)-style objective. This allows a frozen LLM to reconstruct and rephrase the original textual information from the KG.

- Extensive experiments show that GLARE enjoys significant speedup (over $\times 30,000$) while achieving competitive QA performance on complex queries from diverse domains.

## 2 RELATED WORK

**Retrieval Augmented Generation**. Large language models (LLMs) often meet hallucinations and generate unreal responses. Retrieval Augmented Generation (RAG) eases this by retrieving relevant external knowledge for LLMs (Kaddour et al., 2023; Zhang et al., 2023b; Huang et al., 2025). In classic vector-based RAG, external knowledge is split into chunks, converted into embeddings and stored in a vector database. When processing a query, RAG retrieves some relevant chunks based on cosine similarity. Then combines these chunks with the query to create a prompt for generation. However, LLMs struggle to process long contexts due to limited context windows (Tan et al., 2024; Yang et al., 2025). Recent methods try to compress unstructured knowledge base to address this issue, such as the semantic compression method (Mu et al., 2023; Shi et al., 2024) and the pretrained language models method (Xu et al., 2024; Ge et al., 2024). But efficiently compressing textural and structured information, such as on a million-scale KGs, remains underexplored. Our method, GLARE, addresses this problem.

**Knowledge Graph-based RAG**. Knowledge Graph-based RAG uses structured knowledge to enhance the accuracy of LLMs. Existing methods focus on retrieval and generation strategies. Non-parametric retrievers identify relevant nodes and relations efficiently (Gutierrez et al., 2024b; Sarmah et al., 2024), while learning based retrievers excel in matching and retrieval precision (Fang et al., 2024a). Agent-based method uses LLMs to traverse graphs and gather evidence (Sun et al., 2024; Zhang et al., 2024; Ma et al., 2025; Liu et al., 2025a). For generations, topology-aware prompts preserve graph structures for multi-hop reasoning (Wang et al., 2024), and text-based prompts convert graph data into natural language for easier LLM processing (Hu et al., 2024). Although this advantage, KG-based RAG suffers from high space complexity. Recent research, including pruning redundant nodes (Faralli et al., 2018; Jarnac et al., 2023) and using pretrained models to reduce fine-tuning, aims to lower computational costs (Jing et al., 2024; Wang et al., 2021). However, these methods struggle to scale to million-scale KGs. Our method, GLARE, innovatively compresses graph information into a linear vector index, enabling fast retrieval rather than graphs, achieving superior speed and scalability for million-scale KGs.

**Graph Condensation**. Graph condensation makes large graph data into smaller, information-rich subgraphs, while GNNs trained on these smaller graphs achieve the same performance as those trained on original graphs, and this reduces the computational costs of training on GNN (Gao et al., 2025; Fang et al., 2024b). Graph condensation typically has three types. Graph property guided methods create subgraphs that keep key graph features (Jin et al., 2020; Hashemi et al., 2024). Model capability guided methods produce compressed graphs where models perform close to those trained on the original graph (Jin et al., 2022; Xiao et al., 2024). Hybrid methods blend both advantages. Instead of the traditional graph condensation method focuses on GNN training, our method GLARE solves question answering on million-scale KGs by compressing graphs into a linear vector index, achieving efficient retrieval.

## 3 METHOD

### 3.1 OVERVIEW OF GLARE

**Problem Formulation**. We denote the question as $\mathcal{Q}$. For a knowledge graph (KG) $\mathcal{G} = \{(e_i^h, r_i, e_i^t) \mid i = 1, \ldots, N\}$ with over one million entities ($N \geq 1 \times 10^6$) and dense edges between entities, we aim to construct a *vector memory* $\mathcal{M}$, which consists of $M$ vectors, derived from $\mathcal{G}$ such that:

- The reader LLM $f$ achieves comparable QA performance when retrieving from the KG $\mathcal{G}$ or the vector memory $\mathcal{M}$.

- The size of vector memory $\mathcal{M}$ is significantly smaller than the KG $\mathcal{G}$ (i.e., $M \ll N$).

Such a problem can be formulated as the following *Bi-level* objective:

$$\min_{\mathcal{M}} \mathrm{KL}[P(\mathcal{A} \mid g_2^*(\mathcal{M}, \mathcal{Q}), \mathcal{Q}) \mid P(\mathcal{A} \mid g_1^*(\mathcal{G}, \mathcal{Q}), \mathcal{Q})]$$
$$\text{s.t. } g_1^* = \arg\max_{g_1} P(\mathcal{A} \mid g_1(\mathcal{G}, \mathcal{Q}), \mathcal{Q}), \quad g_2^* = \arg\max_{g_2} P(\mathcal{A} \mid g_2(\mathcal{M}, \mathcal{Q}), \mathcal{Q}). \tag{1}$$

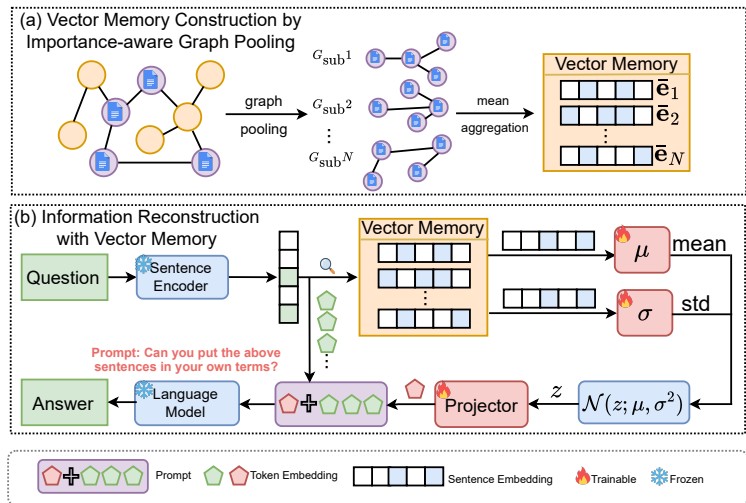

Figure 2: Illustration of the proposed GLARE. In part (a), we use importance-aware graph pooling to generate informative subgraphs $G_{\text{sub}}$ and compress them into vector memory. In part (b), we reconstruct the knowledge graph information from the vector $\bar{\mathbf{e}}$ retrieved from the vector memory.

Here $g_1$ and $g_2$ are retriever models for $\mathcal{G}$ and $\mathcal{M}$, respectively. $P(\cdot|\cdot)$ denotes the distribution of the answer $\mathcal{A}$ generated by the reader large language model (LLM) for question answering conditioned on its input context. By optimizing the bi-level objective in Eq. 1, we can derive an information-intensive vector memory $\mathcal{M}$. Retrieving from the vector memory achieves comparable performance to directly retrieving from the original KG due to the minimization of the Kullback-Leibler (KL) divergence in Eq. 1. However, directly minimizing the objective in Eq. 1 is difficult due to the nesting between the inner and outer optimization problems in bi-level optimization.

To reduce the difficulty of optimization, we reinterpret our goal from a generative perspective. Recall that the size of the vector memory is significantly smaller than the size of KG. Therefore, we aim to project the information of a subgraph $G_{\text{sub}} \in \mathcal{G}$ in KG into a latent variable $z$, which leads to learning the posterior distribution $p(z|G_{\text{sub}})$. In this case, the vector $z$ is latent variable that contains the information in $G_{\text{sub}}$. However, directly learning $p(z|G_{\text{sub}})$ is difficult, which involves learning the posterior of a latent variable. We resort to variational inference and employ a variational distribution $q(z|G_{\text{sub}})$ to approach $p(z|G_{\text{sub}})$, leading to the following objective:

$$\min_{q(z|G_{\text{sub}})} \text{KL}[q(z|G_{\text{sub}})|p(z|G_{\text{sub}})]. \tag{2}$$

By using $p(z|G_{\text{sub}}) = \frac{p(G_{\text{sub}}|z)p(z)}{p(G_{\text{sub}})}$, we reinterpret the KL objective as follows (Kingma et al., 2013) (See Appendix):

$$\log p(G_{\text{sub}}) - \text{KL}[q(z|G_{\text{sub}})|p(z|G_{\text{sub}})] = \mathbb{E}_{z \sim q(z|G_{\text{sub}})} \log p(G_{\text{sub}}|z) - \text{KL}[q(z|G_{\text{sub}})|p(z)]. \tag{3}$$

Since $\log p(G_{\text{sub}})$ is a constant, minimizing Eq. 2 is equal to maximizing the right hand side (R.H.S.) of Eq. 3. Thus, the goal of GLARE is to maximize the following objective:

$$\max_{q(z|G_{\text{sub}})} \mathbb{E}_{z \sim q(z|G_{\text{sub}})} \log p(G_{\text{sub}}|z) - \text{KL}[q(z|G_{\text{sub}})|p(z)]. \tag{4}$$

The first term of Eq. 4 encourages $z$ to be sufficient to reconstruct the information in $G_{sub}$. And the second term of Eq. 4 is to regularize the posterior close to the prior distribution $p(z)$. With Eq. 4, we reduce the load of jointly optimizing the vector memory $\mathcal{M}$ and two retrievers $g_1$ and $g_2$ of the bi-level objective in Eq. 1.

## 3.2 VECTOR MEMORY CONSTRUCTION BY IMPORTANCE-AWARE GRAPH POOLING

The posterior $q(z|G_{\text{sub}})$ naturally offers an efficient manner to construct the vector memory using the information of $G_{\text{sub}}$. The key challenge is how to choose a set of subgraphs that covers the most information in the original KG. This problem is similar to graph pooling, where a trainable

graph pooling module learns a global node assignment matrix. For a KG with $N$ entities, the $(i, j)$ element of the global assignment matrix $A \in \mathbb{R}^{N \times K}$ is the probability of the $i$-th entity belonging to the $j$-th subgraph. However, learning such a global assignment matrix with graph pooling on a million-scale KG is computationally infeasible. Thus, we employ a non-parametric, importance-aware graph pooling method by identifying the influential entities in the KG as the anchor points for graph pooling. We introduce two metrics to select influential entities.

**Centrality**. Centrality (Borgatti, 2005) is a simple and effective metric to select influential entities based on their in-degree and out-degree. The importance score based on centrality is defined as follows:

$$\text{Score}(V_i) = \deg_{\text{in}}(V_i) + \deg_{\text{out}}(V_i), \tag{5}$$

where $\deg_{\text{in}}$ and $\deg_{\text{out}}$ denote the in-degree and out-degree. However, centrality does not consider global information propagation between entities on KGs when evaluating the importance of entities. Thus, we also introduce the PageRank score to evaluate the importance of entities in the KG.

**PageRank**. PageRank (Page et al., 1999) is a global importance metric based on the recursive influence of entities. The motivation of PageRank is that an entity is important if other important entities point to it.

$$\text{Score}(V_i) = \frac{(1 - \alpha)}{N} + \alpha \sum_{V_j \in \text{Nei}(V_i)} \frac{\text{Score}(V_j)}{\deg_{\text{out}}(V_j)} \tag{6}$$

where $\alpha$ is the damping factor used to balance between random jumps and link-based propagation, and $\text{Nei}(V_i)$ denotes the set of neighbor nodes that point to $V_i$.

After identifying the most influential entity $V_i^*$ with the highest scores, we construct the subgraph $G_{\text{sub},i}^*$ around 1-hop neighborhood around $V_i^*$. Then we remove $G_{\text{sub},i}^*$ from the original KG $\mathcal{G}$ to avoid overlap and identify the next most influential entity $V_{i+1}^*$. This process repeats for $M$ iterations and yield $M$ subgraphs $\{G_{\text{sub},i}^* | i = 1, \cdots, M\}$ Such an iterative scheme ensures broader coverage of the original KG with the identified subgraph while reducing redundancy among them.

**Pooled Subgraph Embedding**. After obtaining $M$ subgraphs $\{G_{\text{sub},i}^* | i = 1, \cdots, M\}$ by identifying $M$ influential nodes, we use graph pooling to pool subgraphs into the compact subgraph embeddings. We first employ a pretrained Sentence-Bert (Reimers & Gurevych, 2019) as a text encoder to encode the entities in a KG into entity embeddings. Then we employ the mean aggregation to pool the subgraph into an embedding:

$$\bar{\mathbf{e}}_i = \frac{1}{|G_{\text{sub},i}^*|} \sum_{V_j \in G_{\text{sub},i}^*} \text{Enc}(V_j), \tag{7}$$

where $\text{Enc}(\cdot)$ denotes the text encoder. $|G_{\text{sub},i}^*|$ denotes the volume of $G_{\text{sub},i}^*$. $\bar{\mathbf{e}}_i$ is the pooled subgraph embedding of $G_{\text{sub},i}^*$. Notice that one can employ a more powerful text encoder and advanced graph pooling methods to produce more expressive subgraph embeddings. We focus on the principled framework of GLARE and leave the exploration of the design space for future work.

**Vector Memory Construction**. The resulting $M$ pooled subgraph embeddings construct the vector memory $\mathcal{M} = \{\bar{\mathbf{e}}_1, \ldots, \bar{\mathbf{e}}_M\}$. Each vector in $\mathcal{M}$ is knowledge-intensive as it potentially preserves the information of all the entities in a subgraph. Thus, the vector memory is a compression of the original KG. When the reader LLM receives a query, retrieving from this vector memory is much more efficient than retrieving from KG.

### 3.3 Information Reconstruction with Vector Memory

After constructing the vector memory $\mathcal{M}$, GLARE optimizes the objective in Eq. 4 to decode the messages in $\mathcal{M}$ back into the original information stored in the KG $\mathcal{G}$. Since the vector $\bar{\mathbf{e}}$ in $\mathcal{M}$ is a real-valued vector obtained from $G_{\text{sub}}$ through a deterministic graph pooling process, we have $q(z|G_{\text{sub}}) = q(z|\bar{\mathbf{e}})$. Then we parameterize $q(z|\bar{\mathbf{e}}) = \mathcal{N}(z; \mu, \sigma^2)$ where $\mathcal{N}(z; \mu, \sigma^2)$ is a Gaussian distribution with learnable mean and variance. By setting the prior distribution $p(z)$ as standard Gaussian distribution $\mathcal{N}(z; 0, I)$, we can use the reparametrization trick (Kingma et al., 2013) and covert the second term in Eq. 4 into the following loss:

$$\mathcal{L}_{\text{KL}}(\phi, \theta) = \text{KL}[\mathcal{N}(z; \mu, \sigma^2) \,|\, \mathcal{N}(z; 0, I)], \quad \mu = f_\theta(\bar{\mathbf{e}}), \quad \log \sigma^2 = f_\phi(\bar{\mathbf{e}}). \tag{8}$$

Table 1: Data statistics of STaRK (Wu et al., 2024).

| Dataset | Entity type | Relation type | Avg. degree | Entities | Relations | Tokens |
|---------|-------------|---------------|-------------|----------|-----------|--------|
| STARK-AMAZON | 4 | 5 | 18.2 | 1,035,542 | 9,443,802 | 592,067,882 |
| STARK-MAG | 4 | 4 | 43.5 | 1,872,968 | 39,802,116 | 212,602,571 |
| STARK-PRIME | 10 | 18 | 125.2 | 129,375 | 8,100,498 | 31,844,769 |

Here $f_\theta$ and $f_\phi$ are two Multi-layer Perceptron (MLP). The first term of Eq. 4 encourages the latent variable $z$ to sufficiently reconstruct the information in $G_{\text{sub}}$. We formulate this as a paraphrase reconstruction task. That is, the reader LLM is prompted to "explain" the textual information of the entities in $G_{\text{sub}}$ using $z \sim \mathcal{N}(z; \mu, \sigma^2)$. A trainable projector $\text{Proj}_\varphi$ with parameter $\varphi$ maps $z$ into the token space of **a frozen reader LLM**. Thus, the first term of Eq. 4 is converted to the following loss:

$$\mathcal{L}_{\text{Rec}}(\varphi) = \frac{1}{|G^*_{sub}|} \mathrm{E}_{V_i \in G^*_{\text{sub}}} \mathrm{E}_{z \sim \mathcal{N}(z; \mu, \sigma^2)} - \log P(V_i \mid \textit{prompt}, \text{Proj}_\varphi(z)). \tag{9}$$

Here *prompt* is a text prompt to instruct the reader LLM to reconstruct the textual information using $z$ and the prompt could be found in Figure 2. $\mathcal{L}_{\text{Rec}}$ is computed using negative log-likelihood (NLL) loss between the auto-aggressive token generation of the reader LLM and the ground-truth token sequence of $V_i$'s textual information. To preserve the graph structure, we further introduce a structure-preserving loss:

$$\mathcal{L}_{\text{struct}} = \|ZZ^T - A\|_F^2, \tag{10}$$

where $Z$ denotes the latent variables sampled for each node and $A$ is the adjacency matrix of the corresponding subgraph. This regularization encourages the latent representation to retain topological information of the KG. Thus, the overall loss of the information reconstruction is:

$$\mathcal{L}(\phi, \theta, \varphi) = \mathcal{L}_{\text{Rec}}(\varphi) + \mathcal{L}_{\text{KL}}(\phi, \theta) + \mathcal{L}_{\text{struct}}, \tag{11}$$

### 3.4 LIGHTWEIGHT RETRIEVAL WITH GLARE

Through importance-aware graph pooling and information reconstruction, GLARE compresses the KG with millions of entities into an information-intensive vector memory. Unlike traditional KG-based RAG methods that directly retrieve from a vast graph space, GLARE achieves a lightweight retrieval from vector memory to facilitate question answering, only involving $O(n)$ computational complexity as the size of the vector memory $\mathcal{M}$ increases. Thus, GLARE can scale to a million-scale KG, while traditional KG-based RAG methods only handle KGs with thousands of entities. The retrieval process of GLARE takes the following steps:

**Step 1**. A pretrained LM encodes the input question $\mathcal{Q}$ into the question embedding.

**Step 2**. Retrieve $\bar{\mathbf{e}}$ from $\mathcal{M}$ with the highest vector similarity to the question embedding.

**Step 3**. Compute $\mu = f_\theta(\bar{\mathbf{e}})$ and $\sigma = \sqrt{e^{f_\phi(\bar{\mathbf{e}})}}$ based on Eq. 8. Sample $z \sim \mathcal{N}(z; \mu, \sigma)$ using parameterization trick: $z = \mu + \sigma \cdot \epsilon$ where $\epsilon \sim \mathcal{N}(0, I)$.

**Step 4**. The reader LLM generates answer $\mathcal{A}$ with $z$ by $P(\mathcal{A}|\text{Proj}_\varphi(z))$.

## 4 EXPERIMENT

### 4.1 DATASET AND METRICS

**Dataset**. We use the STaRK (Wu et al., 2024) benchmark to evaluate the performance of GLARE in efficient retrieval for QA. STaRK is a recently proposed benchmark consisting of three structured knowledge graphs across different domains. Detailed statistics on the number of entities, relations, tokens, and average degree can be found in Table 1. Below is a brief introduction of the three sub-datasets:

- **STaRK-AMAZON** is built from an e-commerce platform. It includes entity types such as products, brands, colors, and categories. The textual information mainly comes from product descriptions, customer reviews, and Q&A data.

Table 2: Overall performance comparison of different methods across the three STaRK sub-datasets (Amazon, Prime, and Mag) under four evaluation metrics. Bold = best, underline = second-best.

| Method | STaRK-AMAZON | | | | STaRK-PRIME | | | | STaRK-MAG | | | |
|---|---|---|---|---|---|---|---|---|---|---|---|---|
| | F1 score | BERT score | Human eval | LLM score | F1 score | BERT score | Human eval | LLM score | F1 score | BERT score | Human eval | LLM score |
| Without RAG | 1.57 | 45.71 | 29.59 | 12.16 | 0.50 | 42.42 | 23.62 | 6.46 | 0.46 | 41.83 | 16.45 | 8.08 |
| xRAG | 19.13 | 54.38 | 39.52 | 7.36 | 5.23 | **53.75** | 30.64 | 11.52 | 20.31 | 41.34 | 40.30 | 6.13 |
| RAG | 17.07 | 48.38 | 45.07 | **16.70** | 5.46 | 40.53 | 47.47 | 16.94 | 12.67 | 49.84 | 24.32 | 0.74 |
| Noise | 20.90 | 54.81 | 40.55 | 8.26 | **7.03** | 45.49 | 28.53 | 12.61 | 19.86 | 53.48 | 28.69 | 8.04 |
| SubgraphRAG | 12.68 | 48.92 | 27.40 | 2.97 | 2.81 | 37.99 | 29.90 | 3.98 | 15.50 | 51.04 | 16.67 | 2.18 |
| PCST | 10.15 | 47.65 | 21.78 | 1.24 | 2.95 | 38.26 | 32.89 | 6.96 | 14.44 | 50.53 | 21.19 | 1.02 |
| ToG+LLaMA3 | 16.70 | 51.95 | 47.95 | 4.12 | 4.18 | 41.76 | 38.69 | 9.55 | 12.56 | 50.23 | 21.63 | 0.56 |
| ToG+GPT-4o | 23.17 | 49.82 | 44.40 | 8.15 | 4.77 | 42.24 | 41.47 | 14.10 | 0.45 | 42.80 | 7.90 | 2.13 |
| **GLARE-Deg** | **23.61** | **55.97** | 53.29 | 12.18 | 5.86 | 46.58 | **47.69** | 16.82 | 20.49 | 53.65 | 44.31 | **8.92** |
| **GLARE-PgRank** | 22.18 | 53.59 | **54.82** | 12.79 | 5.27 | 43.72 | 45.83 | **17.03** | **20.83** | **54.81** | **44.86** | 8.57 |

- **STaRK-MAG** is constructed based on the Microsoft Academic Graph. It contains entities such as papers, authors, institutions, and research fields, and includes text from paper titles, abstracts, and citation data. This dataset supports paper search and question answering.

- **STaRK-PRIME** is from the biomedical knowledge graph PrimeKG. It covers ten types of entities, including diseases, genes, drugs, and pathways, and includes eighteen types of relations. It combines rich textual descriptions and is designed for complex queries in the domain of precision medicine.

Although STaRK was originally designed as a retrieval benchmark, it can be naturally adapted into a QA dataset. Each entry contains a query and a set of answer nodes from a structured knowledge base. Since these nodes represent entities with names or titles, they can serve as the answer. For example, in STARK-AMAZON, the retrieved texts are products identified by their names; in STARK-MAG, the results are academic papers with clear titles; and in STARK-PRIME, the answers include biomedical concepts such as drug names, disease names, or gene identifiers. By extracting the name or title of each matched text, STaRK can be effectively used as a multi-answer QA dataset.

**Metrics**. We employ the following metrics to evaluate the retrieval performance.

- **F1 score** This metric computes the harmonic mean of precision and recall over token overlaps between generation answers and ground truth. As it doesn't rely on trained models, it's ideal for rapid and consistent answer quality assessment (Lewis et al., 2020; Guu et al., 2020; Karpukhin et al., 2020; Chen et al., 2017; Izacard & Grave, 2021).

- **Bert Score** We evaluate the semantic embedding similarity between generated and ground truth from pre-trained language models(deberta-xlarge-mnli) (He et al., 2021). But it may overestimate answers that are semantically similar but factually incorrect (Wang et al., 2023; Zhang et al., 2019; Sellam et al., 2020; Zhao et al., 2020; Yuan et al., 2021).

- **Human eval** We randomly sample 300 pairs of question-answer from generated answers and manually score them on a scale of 0 to 100 based on a unified evaluation rubric. Each answer is independently evaluated by two reviewers, and we take the average score as the final result. We evaluate the answer based on accuracy, completeness, and clarity, and it is the most reliable evaluation method. The evaluation rubric is available in A.3.

- **LLM Score** We design specific prompts, including the question, ground truth, and generated answer, and use a LLM (Qwen2.5-Max) (Qwen et al., 2025) to obtain a score between 0 and 100. This method evaluates responses, accommodates different expressions, and shows strong robustness and judging ability in automatic evaluation (Dubois et al., 2023; Dettmers et al., 2023). More details about the prompt can be found in A.4.

Table 3: Overall average performance comparison of different methods across the STaRK.

| Method | F1 score | BERT score | Human eval | LLM score |
|---|---|---|---|---|
| Without RAG | 0.84 | 43.32 | 23.22 | 8.90 |
| xRAG | 14.89 | 49.82 | 36.82 | 8.34 |
| RAG | 11.73 | 46.25 | 38.95 | 11.46 |
| Noise | 15.93 | 51.26 | 32.59 | 9.64 |
| SubgraphRAG | 10.33 | 45.98 | 24.66 | 3.04 |
| PCST | 9.18 | 45.48 | 25.29 | 3.07 |
| ToG+LLaMA3 | 11.15 | 47.98 | 36.09 | 4.74 |
| ToG+GPT-4o | 9.46 | 44.95 | 31.26 | 8.13 |
| **GLARE-Deg** | **16.65** | **52.07** | 48.43 | 12.64 |
| **GLARE-PgRank** | 16.09 | 50.71 | **48.50** | **12.80** |

Table 4: Retrieval latency, relative speedup, and LLM score.

| Method | Time (ms) | Relative Speed | LLM score |
|---|---|---|---|
| GLARE | 0.046 | 1× | 12.80 |
| xRAG | 0.096 | 2.09× | 8.34 |
| RAG | 0.293 | 6.37× | 11.46 |
| Noise | / | / | 9.64 |
| Without RAG | / | / | 8.90 |
| SubgraphRAG | 94 | 2043× | 3.04 |
| RoG | 1283 | 27891× | 4.35 |
| ToG+LLaMA | 1035 | 22500× | 4.74 |
| ToG+GPT-4o | 1449 | 31500× | 8.13 |

## 4.2 BASELINES

We compare GLARE with a diverse set of baselines, organized into four categories based on their strategies for representing and using KG information.

**Standard RAG**. Standard RAG methods serve as a basic benchmark for evaluating external knowledge usage. **Without RAG** directly generates answers without any retrieval, as a pre-LLM baseline. **RAG** flattens the KG into a linear document collection and retrieves nodes via embedding similarity.

**KG based RAG Methods**. We selected representative graph structured methods as baselines, all based on the use of KG relational and topological information. **ToG(Sun et al., 2024)+LLaMA3** and **ToG+GPT-4o** are agent-based methods that use strong LLMs to iteratively retrieve from the KG. **SubgraphRAG** (Li et al., 2025a) retrieves relevant triplets using a learnable module. **PCST** (Archer et al., 2011) solves a graph optimization problem to construct a compact subgraph per query. All these methods retrieve knowledge from the KG. However, since these methods cannot scale to million-scale KGs, we adopt a hybrid setting to ensure computational feasibility. We first identify the seed entity using dense retrieval and deploy graph RAG methods within the 2-hop ego-graph centered at the seed entity.

**Standard Compression Methods**. To further balance efficiency and retrieval quality, we test methods that avoid structured retrieval. **xRAG**(Cheng et al., 2024) flattens the knowledge graph and retrieves similar nodes via embeddings, then compresses their textual contents into a token as a soft prompt using a trainable projector. **Noise** directly samples a latent vector from a Gaussian distribution without retrieving anything, feeding it into the projector, which uses the generated vector as a soft prompt. This baseline evaluates whether the projector alone performs well without any input.

## 4.3 MAIN RESULTS

**Overall Performance Comparison**. From Table 2 and Table 3, GLARE achieves strong performance while maintaining high compression and fast retrieval speed. Specifically, GLARE-PgRank attains the highest score in LLM score and Human eval, while ranking second in F1 score. GLARE-Deg achieves the best results in F1 score and BERT score, and ranks second in Human eval and LLM score, showing GLARE's strength in knowledge-intensive tasks. RAG ranks third in LLM score and Human eval, outperforming other baselines, which indicates that flattening the graph for semantic retrieval remains effective. Latent compression methods show mixed results. Noise performs better than xRAG, demonstrating the projector's strong ability in generation. ToG+LLaMA3 and ToG+GPT-4o perform well on Human eval but suffer from LLM score, reflecting that they can use LLM reasoning through multi-hop graph traversal to ease the retrieval challenge. Although our focus is on million-scale KGs, unlike small benchmarks such as WebQSP and CWQ (only thousands of entities), we also report results on them to examine generalization, which are provided in A.1.

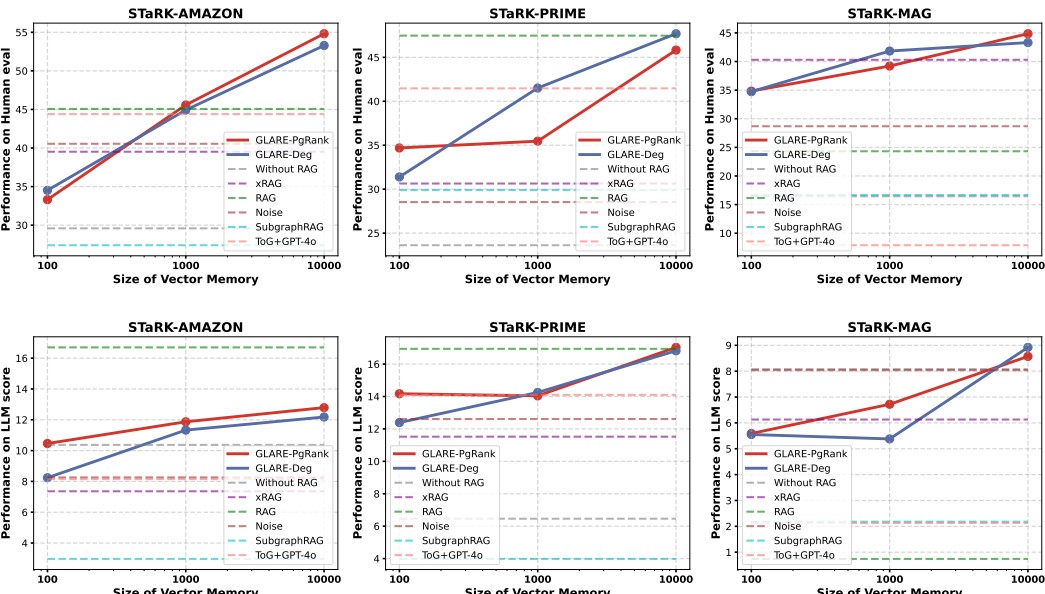

Figure 3: Performance comparison of Glare with different size of vector memory across STaRK-AMAZON, STaRK-PRIME and STaRK-MAG. The top row presents Human eval results, and the bottom row reports LLM score. The red and blue lines represent GLARE-Deg and GLARE-PgRank, respectively. Colored dashed lines indicate the performance of baseline methods.

## 4.4 DISCUSSION

**Retrieval Time Cost Analysis**. To further evaluate the response efficiency of different methods, we measured the average retrieval time (in milliseconds) during the knowledge retrieval stage and compared each method's relative speed to our proposed GLARE. The results are shown in Table 4.

GLARE significantly outperforms all other methods in retrieval efficiency and is used as the baseline (1×). Compression-based methods, such as xRAG and RAG, are relatively fast but still 10× slower than GLARE. By contrast, traditional graph-based methods are 2000× slower because they require costly subgraph selection, while agent-based methods are 30000× slower due to multi-round reasoning and interaction, making both categories unsuitable for large-scale KGs. In summary, GLARE achieves much faster retrieval speeds while maintaining high generation quality, making it particularly suitable for large-scale knowledge graph QA tasks.

**Influence of Vector Memory Size**. We study how vector memory size influences GLARE's performance. Figure 3 reports results on three STaRK datasets (Amazon, Prime, Mag) by using Human eval and LLM score. At a 100× compression ratio (10,000 vectors), GLARE outperforms all baseline methods on both metrics across all datasets. This shows that even with substantial compression, GLARE retains the essential knowledge needed for high-quality answer generation. At an extreme compression ratio of 10,000× (100 vectors), GLARE still surpasses some baselines, indicating that despite inevitable information loss, the projector can preserve crucial structures for QA. Finally, performance grows linearly with larger vector memory, highlighting the scalability and robustness of our method for large-scale knowledge graphs.

## 5 CONCLUSION

We propose GLARE, a novel method for handling a million-scale knowledge graph RAG. By compressing the knowledge graph into a high-density vector memory, the retrieval model only needs linear-time complexity to retrieve in the vector memory, eliminating the need for extensive graph space retrieval. This enables fast and accurate retrieval on a million-scale KGs. Extensive testing on the STaRK dataset demonstrates that our method not only matches or surpasses the performance of KG-based RAG baselines but also achieves a 100× compression rate and a retrieval speed 30,000× faster than KG-based RAG. These results highlight GLARE's advantages in high compression and fast retrieval speed for million-scale KGs.

## 6 REPRODUCIBILITY STATEMENT

The method details are described in Section 3 and Section 4. In addition, the Appendix provides the details of the experimental environment, hyperparameter configurations, and additional results. All datasets used in our experiments are publicly available.

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

# A APPENDIX

## A.1 PERFORMANCE ON OTHER DATASETS

Beyond STaRK, we evaluate GLARE on smaller benchmarks, WebQSP and CWQ, to examine generalization. While these datasets contain only thousands of entities and are less suitable for testing scalability, STaRK's million-scale better reflect real-world challenges. Nevertheless, we report results on WebQSP and CWQ (Tables 5 and 6) to show that both GLARE variants consistently outperform Subgraph and RoG, confirming that GLARE's graph-less design generalizes well beyond large-scale settings.

Table 5: Performance on WebQSP dataset.

| Method | F1 Score | BERT Score | LLM Score |
|---|---|---|---|
| GLARE-Deg | 74.43 | 63.15 | 68.67 |
| GLARE-PgRank | 73.28 | 64.53 | 67.24 |
| Subgraph | 64.10 | / | / |
| RoG | 70.80 | / | / |

Table 6: Performance on CWQ dataset.

| Method | F1 Score | BERT Score | LLM Score |
|---|---|---|---|
| GLARE-Deg | 62.53 | 56.26 | 60.26 |
| GLARE-PgRank | 63.61 | 54.89 | 57.13 |
| Subgraph | 47.10 | / | / |
| RoG | 56.20 | / | / |

## A.2 INSTRUCTIONS FOR READER LLM TO RECONSTRUCT THE TEXTUAL INFORMATION

The list of prompts for the "explain" instruction is used by the reader LLM to reconstruct the textual information of entities in $G_{\text{sub}}$. These prompts are shown in Table 7. They offer natural language variations while preserving the same intent.

Table 7: Instructions for reader LLM to reconstruct the textual information where [X] refers to the $\text{Proj}_{\varphi}(z)$ and [D] refers to the document D.

| # | Template Sentence |
|---|---|
| 1 | [X] conveys the same underlying message as [D]. |
| 2 | Put simply, background: [X] is another formulation of [D]. |
| 3 | [X] and [D] essentially communicate the same idea. |
| 4 | Restated differently, background: [X] translates to: [D]. |
| 5 | [X] represents a rewording of the concept found in [D]. |
| 6 | To rephrase, background: [X] can be understood as: [D]. |
| 7 | [X] is merely a different way of expressing what [D] says. |
| 8 | If you interpret background: [X], you'll arrive at: [D]. |
| 9 | [X] ultimately amounts to the same meaning as [D]. |
| 10 | The meaning behind background: [X] can be mirrored in: [D]. |
| 11 | [X] expresses the same notion that's encapsulated in [D]. |
| 12 | [X] can be paraphrased straightforwardly as: [D]. |
| 13 | Boiled down, [X] aligns closely with what [D] conveys. |
| 14 | In clearer terms, [X] reflects the same content as [D]. |
| 15 | There's no real difference in meaning between [X] and [D]. |

## A.3 EVALUATION RUBRIC FOR HUMAN EVAL

The detailed evaluation rubric for scoring LLM outputs based on accuracy, completeness, and clarity in Human eval is shown in Table 8.

Table 8: Evaluation rubric for Human eval.

| Evaluation Dimension | Scoring Criteria |
|---|---|
| **Content Accuracy (0–70 pts)** | **60–70**: Fully accurate – all key concepts/entities match the correct answer.
**30–59**: Partially accurate – some concepts match, others are missing or incorrect.
**0–29**: Mostly or completely inaccurate – incorrect or unrelated concepts. |
| **Completeness (0–20 pts)** | **15–20**: Complete – covers all key points present in the correct answer.
**5–14**: Partially complete – some key points missing.
**0–4**: Incomplete – most or all key points missing. |
| **Clarity of Expression (0–10 pts)** | **8–10**: Clear – well-expressed and easy to follow.
**4–7**: Generally clear – understandable but with minor issues.
**0–3**: Confusing – poorly expressed or hard to interpret. |

## A.4 PROMPTS FOR LLM SCORE

The specific prompts used by LLMs to evaluate LLMs performance for LLM Score, incorporating the question, ground truth, and generated answer, are detailed are following.

---

**LLM Score Evaluation Prompt**

The question is: { }
The correct answer: { }
The student's answer: { }

You are an evaluation module tasked with assessing the alignment between a model-generated answer and the correct reference answer, given a specific question. This is a subjective evaluation, so while the wording in the generated response may vary, correctness depends on whether it refers to the same entities or concepts as the reference. If the content refers to different entities or concepts, it should be deemed incorrect. The score should be determined based on the proportion of correctly mentioned items relative to the total expected items, to reflect the overall semantic accuracy of the response.

---

## A.5 IMPLEMENTATION DETAILS

We used the language model Mistral-7B, a 7.3B parameter open-weight model which has a strong performance across a wide range of nature language tasks. In our evaluation, for Standard RAG and KG-based RAG methods, we directly concatenate the retrieved content with the query to form the prompt, and then passed to the Mistral-7B model to generate answers. For Standard Compression methods and our method Glare, which involve using token-space soft prompts as the retrieval content, we fine-tune Mistral-7B and use the instruction tuning variant of the model to produce answers.

Owing to efficiency constraints, we do not perform on-the-fly retrieval. All baselines and our method pre-constructed a retrieval index and using their own retrieval method to evaluation on QA tasks.

We employ the embedding model SFR-Embedding-Mistral, a text embedding model that achieves top performance on the MTEB benchmark by leveraging multi-task fine-tuning across retrieval, clustering, and classification tasks. All operations involving text-to-embedding conversion are transformed by using the SFR-Embedding-Mistral model. All experiments are conducted using 8 Nvidia A100 GPUs.

In Table 9, we list the hyperparameters used for training our method GLARE on the paraphrase reconstruction task.

Table 9: Hyperparameters for paraphrase reconstruction pretraining of GLARE

| Hyperparameter | Assignment |
|---|---|
| Optimizer | AdamW |
| Learning rate | 1e-5 |
| LR scheduler type | Linear |
| Warmup ratio | 0.03 |
| Weight decay | 0.1 |
| Epochs | 1 |
| Batch size | 6 |
| Gradient accumulation steps | 8 |
| Max sequence length | 336 |
| Max train samples | 20,000 |
| Flash attention | True |
| Gradient checkpointing | True |
| KL loss weight ($\alpha_{\text{KL}}$) | 0.1 |
| NLL loss weight ($\alpha_{\text{NLL}}$) | 1.0 |
| Number of GPUs | 8 |

## A.6 DERIVATION OF ELBO FROM KL DIVERGENCE

In this section, we derives the transformation from the KL divergence in Eq. 2 to the evidence lower bound (ELBO) in Eq. 3 for GLARE.

In GLARE, our goal is to use a variational distribution $q(z|G_{\text{sub}})$ to approximate the posterior $p(z|G_{\text{sub}})$ of a latent variable $z$ given a knowledge graph subgraph $G_{\text{sub}}$. So it's to minimize:

$$\min_{q(z|G_{\text{sub}})} \text{KL}[q(z|G_{\text{sub}})|p(z|G_{\text{sub}})], \tag{12}$$

where $\text{KL}[q(z|G_{\text{sub}})|p(z|G_{\text{sub}})] = \int q(z|G_{\text{sub}}) \log \frac{q(z|G_{\text{sub}})}{p(z|G_{\text{sub}})} \, dz$.

By using Bayes' theorem, the posterior is:

$$p(z|G_{\text{sub}}) = \frac{p(G_{\text{sub}}|z)p(z)}{p(G_{\text{sub}})}. \tag{13}$$

Substitute it into the KL divergence:

$$\text{KL}[q(z|G_{\text{sub}})|p(z|G_{\text{sub}})] = \int q(z|G_{\text{sub}}) \log \frac{q(z|G_{\text{sub}})}{\frac{p(G_{\text{sub}}|z)p(z)}{p(G_{\text{sub}})}} \, dz$$

$$= \int q(z|G_{\text{sub}}) \left[ \log q(z|G_{\text{sub}}) + \log p(G_{\text{sub}}) - \log p(G_{\text{sub}}|z) - \log p(z) \right] \, dz$$

$$= \int q(z|G_{\text{sub}}) \log \frac{q(z|G_{\text{sub}})}{p(z)} \, dz - \int q(z|G_{\text{sub}}) \log p(G_{\text{sub}}|z) \, dz + \int q(z|G_{\text{sub}}) \log p(G_{\text{sub}}) \, dz \tag{14}$$

The first term is:
$$\int q(z|G_{\text{sub}}) \log \frac{q(z|G_{\text{sub}})}{p(z)} \, dz = \text{KL}[q(z|G_{\text{sub}})|p(z)], \tag{15}$$

The second term is:
$$\int q(z|G_{\text{sub}}) \log p(G_{\text{sub}}|z) \, dz = \text{E}_{z \sim q(z|G_{\text{sub}})} \left[ \log p(G_{\text{sub}}|z) \right], \tag{16}$$

And since $p(G_{\text{sub}})$ is constant, the third term is:
$$\int q(z|G_{\text{sub}}) \log p(G_{\text{sub}}) \, dz = \log p(G_{\text{sub}}) \int q(z|G_{\text{sub}}) \, dz = \log p(G_{\text{sub}}). \tag{17}$$

Thus:
$$\text{KL}[q(z|G_{\text{sub}})|p(z|G_{\text{sub}})] = \text{KL}[q(z|G_{\text{sub}})|p(z)] - \text{E}_{z \sim q(z|G_{\text{sub}})} \left[ \log p(G_{\text{sub}}|z) \right] + \log p(G_{\text{sub}}), \tag{18}$$

$$\log p(G_{\text{sub}}) - \text{KL}[q(z|G_{\text{sub}})|p(z|G_{\text{sub}})] = \text{E}_{z \sim q(z|G_{\text{sub}})} \left[ \log p(G_{\text{sub}}|z) \right] - \text{KL}[q(z|G_{\text{sub}})|p(z)]. \tag{19}$$

This is ELBO, as shown in Eq. 3. Minimizing $\text{KL}[q(z|G_{\text{sub}})|p(z|G_{\text{sub}})]$ is equal to maximizing the right side $\text{E}_{z \sim q(z|G_{\text{sub}})} \left[ \log p(G_{\text{sub}}|z) \right] - \text{KL}[q(z|G_{\text{sub}})|p(z)]$

This process enables GLARE to optimize the variational distribution for subgraph compression.

### A.7   LLM USAGE

We used large language models solely as a general-purpose writing assist tool for correcting grammar and improving readability. LLMs were not involved in the research idea, method design, experiments, analysis, or conclusion.

