# OpenReview forum: "GLARE: Towards Graph-less Retrieval for Retrieval Augmented Generation on Million-scale Knowledge Graphs"
_ICLR.cc/2026/Conference — Submitted to ICLR 2026_

### Official Review · Reviewer_PcSX · 2025-10-29

**Soundness:** 2
**Presentation:** 2
**Contribution:** 3
**Rating:** 4
**Confidence:** 3

**Summary:**

To address high retrieval latency in knowledge-graph-based (KG-based) RAG, this paper introduces GLARE, a KG-based RAG framework that enables fast and accurate information processing over million-scale KGs. Specifically, GLARE compresses large KGs into a vector memory. To preserve critical information, GLARE employs  a non-parametric, importance-aware graph pooling strategy and a VAE-style
projector that reconstructs relational structures from the vector memory. Empirical studies demonstrate the effectiveness and efficiency of the proposed approach.

**Strengths:**

**S1.** The paper proposes a novel methodology for KG-based RAG with intuitive motivations, combining ideas from network science, information retrieval, graph neural networks, and VAEs.

**S2.** Empirical studies demonstrate the effectiveness and efficiency of the proposed approach.

**S3.** The paper is overall easy to follow.

**Weaknesses:**

I'm willing to adjust my rating if some of important issues can be addressed.

**W1.** The proposed approach has inherent limitations in its expressive power.
- The subgraph selection process is fully structure-based, ignoring semantic information. Introducing some clustering-based ideas may help.
- Only one-hop subgraphs are extracted.
- The subgraph pooling process also ignores relational semantic information.

**W2.** This paper only considers evaluations on the STaRK benchmark, while many previous KG-based RAG approaches were developed and evaluated on other datasets. It remains unclear whether the proposed approach is generalizable to those widely adopted datasets. While Appendix A.1 presents some comparisons on WebQSP and CWQ, they remain limited with insufficient baselines. In particular, the STaRK graphs have very few numbers of entity and relation types (<20) compared to KGs like FreeBase and Wikipedia. The questions in other KG question answering datasets can also present different supporting graph structures.

**W3.** The paper still misses some highly relevant papers in related work discussion, such as [1], [2], [3], [4]. Expanding the related work discussion can help readers better assess your contributions. E.g., RoG [1] also does not suffer from KG scalability issues. I notice that the paper does compare against [1] in empirical studies (Table 4-6), but the baseline is not mentioned in the text, which is weird. Some discussions and comparisons in terms of methodology design will also be helpful.

[1] Luo et al. Reasoning on Graphs: Faithful and Interpretable Large Language Model Reasoning. ICLR 2024.

[2] Luo et al. Graph-constrained Reasoning: Faithful Reasoning on Knowledge Graphs with Large Language Models. ICML 2025.

[3] Mavromatis & Karypis. GNN-RAG: Graph Neural Retrieval for Large Language Model Reasoning. ACL Findings 2025.

[4] Chen et al. Plan-on-Graph: Self-Correcting Adaptive Planning of Large Language Model on Knowledge Graphs.

**W4.** Figure 1 claims that the time complexity of retrieval in knowledge graph is $O(2^n)$ while in vector memory is $O(n)$. This contrasting difference is not sufficiently discussed and justified other than the figure caption.

**W5.** Equations and notations.
- Equation (1) is problematic and should take expectations over the sample distributions as you do not construct a vector memory for each question.
- Equations like (2) can be made more clear by using $\phi$ or $\theta$ to denote the parametrized model to be learned.

**W6.** Important experiment details are not sufficiently discussed and justified. E.g., the LLM used for the proposed approach and baselines. The hyperparameters used for the baselines, which can significantly affect the baseline performance, e.g., the number of retrieved triples / paths.

**W7.** The paper misses some ablation studies like the impact of different loss terms in equation (11). E.g., Is VAE really needed compared to AE? To what extent does the structure preserving loss help?

**Questions:**

**Q1.** For the final $Project_\phi$, do you project $z$ into the token space or token embedding space?

---

### Official Review · Reviewer_V2Za · 2025-10-31

**Soundness:** 3
**Presentation:** 3
**Contribution:** 3
**Rating:** 4
**Confidence:** 5

**Summary:**

GLARE compresses a million-scale knowledge graph into a dense vector memory, enabling linear-time retrieval without expensive graph traversals. On the STaRK dataset it matches or exceeds KG-based RAG baselines while achieving roughly 100× compression and ~30,000× faster retrieval, highlighting its strong compression and retrieval-speed advantages for large KGs.

**Strengths:**

1.	Tackles a timely, impactful problem relevant to both industry and research.
2.	Proposes a compact vector-memory representation of the KG to cut retrieval overhead.
3.	Demonstrates dramatic retrieval-speed improvements relative to existing RAG and graph-RAG implementations.

**Weaknesses:**

1.	The core idea resembles prior work but this similarity is not addressed in the manuscript.
2.	Evaluation is limited to the STaRK benchmark, which restricts the generality of the results.
3.	The chosen baselines are not sufficiently representative of related methods.

**Questions:**

1.	The approach appears similar to Niu et al. (ECML PKDD 2023), which uses node and graph memory modules. GLARE seems to rely mainly on a graph-level memory—please clarify distinctions and cite/discuss related methods in the main text.
2.	STaRK alone is insufficient to establish robustness. Please add additional benchmarks to better demonstrate comparative performance.
3.	Include more representative baselines (e.g., GraphRAG, LightRAG, RAPTOR, and other relevant methods) to strengthen comparative claims.

---

### Official Review · Reviewer_WhFN · 2025-11-01

**Soundness:** 2
**Presentation:** 3
**Contribution:** 2
**Rating:** 4
**Confidence:** 4

**Summary:**

This paper introduces GLARE for efficient GraphRAG on large-scale KGs. GLARE embeds subgraphs in KGs and builds a vector memory for the embeddings of the subgraphs. During query time, graph traversal is avoided; instead, subgraph embeddings are retrieved according to the query embedding. To let the LLM understand the retrieved subgraph embeddings, a VAE-style projector is trained, which maps the message in vector memory into the token space of the frozen LLM. Experiments on the STaRK benchmark show the effectiveness of GLARE.

**Strengths:**

- The proposed method is fast due to retrieving in the embedding space, but not directly over graph data.
- If there are no similar methods previously, the proposed projector is interesting, which can let LLMs understand subgraph embeddings.
- It shows great capability even with significant compression.

**Weaknesses:**

- The proposed method introduces information loss & locality bias. In some cases, this could improve efficiency while keeping accuracy, but it will also cause issues in cases, e.g., requiring long-hop reasoning. In the meantime, long-tail information in KGs would also be easily lost.
    - I think this could be the main weakness of the method, so I believe it would be better to evaluate GLARE on long-hop QAs.
- I might miss this in the paper, but how is the projector trained, what's the exact arch of the projector, and what's the cost to train the projector?
- How do different anchor policies affect results? If pooling with multiple hop info, how would this affect results?

**Questions:**

See above.

---

### Official Review · Reviewer_pceQ · 2025-11-01

**Soundness:** 2
**Presentation:** 3
**Contribution:** 3
**Rating:** 4
**Confidence:** 4

**Summary:**

This paper proposes GLARE, a "graph-less" RAG framework intended to address retrieval latency on million-scale KG. The method compresses the KG into a compact vector memory using an importance-aware pooling strategy and a VAE-style projector to reconstruct information. The stated goal is to accelerate retrieval from graph space while maintaining question-answering quality. Experiments on STaRK show a significant latency reduction with moderate accuracy.

**Strengths:**

- The method achieves a significant retrieval speedup on million-scale KGs compared to graph-traversal baselines like ToG+GPT-4o.
- The "graph-less" approach, which compresses the KG into a vector memory, is a sound contribution for avoiding costly graph traversal.
- The paper is well-structured and the core idea is clearly presented.

**Weaknesses:**

- The claimed performance gains are not convincing. The method struggles to consistently outperform the standard (and much simpler) RAG baseline, and in some cases, performs worse.
- Absence of ablation experiments. The paper fails to properly analyze the contribution of its core components (the pooling strategy and the VAE projector) against reasonable alternatives.

**Questions:**

- Q1. As shown in Table 2 and Figure 3, GLARE does not perform significantly better than other models in terms of non-human evaluation metrics. Given the significant added complexity of the VAE-based compression, could you elaborate on the practical justification for this method over  much simpler RAG baselines?
- Q2. The contributions of the **Vector Memory Construction** and **Information Reconstruction** modules are not quantified. To assess the impact of your specific design choices, please provide ablation experiments. For instance:
    1. How does the 'importance-aware' pooling (Centrality/PageRank)  compare against both a simple random selection baseline and a more advanced method like SubgraphRAG’s DDE [1]?
    2. How critical is the VAE-style projector? What is the performance when swapping it with a deterministic projector, such as those used in G-Retriever [2] or GraphLLM [3]?

I will consider raising the final rating score if you resolve the above issues.

[1] Retrieval or reasoning: The roles of graphs and large language models in efficient knowledge-graph-based retrieval-augmented generation

[2] G-Retriever: Retrieval-Augmented Generation for Textual Graph Understanding and Question Answering

[3] GraphLLM: Boosting Graph Reasoning Ability of Large Language Model

---

### Meta-Review · Area_Chair_8roS · 2025-12-13

**Summary:**

The paper proposes incorporating KG via training an encoder representation. This can lead to potentially improved efficiency in comparison to traditional GraphRAG approaches. However, reviewers clearly point out the ways with which the paper can be improved. 1) Some of the claims need to be verified with further experimentation, in particular the 30Kx improvement in scale 2) Missing of the literature and comparison to the work highlighted in PcSX's review. 3) The encoder representation needs to be also discussed wrt prior art trainig GNN embeddings.

Without a rebuttal, none of the reviewers would change their score. As such, this paper is not ready for acceptance.

**Reviewer Concerns:**

None.

**Reviewer Scores:**

None of the reviewer scores would change given the lack of a rebuttal.

---

### Decision · Program_Chairs · 2026-01-26

Reject